# The impact of wearing compression hosiery and the use of assistive products for donning and doffing: A descriptive qualitative study into user experiences

Edith Hagedoren—Meuwissen[1,2]*, Uta Roentgen[1], Sandra Zwakhalen[2,3], Loek van der Heide[1], Marie Josee van Rijn[4], Ramon Daniëls[1,2]

1 Research Centre on Assistive Technology in Care, Zuyd University of Applied Sciences, Heerlen, The Netherlands, 2 Department of Health Services Research, Care and Public Health Research Institute, Maastricht University, Maastricht, The Netherlands, 3 Academy of Nursing, Zuyd University of Applied Sciences, Heerlen, The Netherlands, 4 Department of Vascular Surgery, Erasmus MC, Rotterdam, The Netherlands

* edith.hagedoren@zuyd.nl

## Abstract

This qualitative study aimed to describe users' experiences and needs related to wearing, donning, and doffing compression hosiery, and the provision process of compression hosiery and associated assistive products for donning and doffing. Adults who have been advised to wear compression hosiery participated in semi-structured interviews. Existing frameworks about the provision process and acceptance of assistive technology guided the topic list. The interviews were analyzed using directed content analysis. After 19 interviews, data saturation was reached. Three main themes were revealed: (1) becoming and being a compression hosiery wearer (or not), (2) wearing compression hosiery, and (3) donning and doffing compression hosiery. In cases of acute conditions, the participants reported feeling overwhelmed when they were first prescribed compression hosiery. In contrast, those with long-term complaints felt relieved. Participants considered advantages and disadvantages and then decided whether to wear compression hosiery. Despite mentioned beneficial effects from wearing compression hosiery, participants had to cope with a range of discomforts, including pinching, straining, sagging, and heat. Additionally, participants had difficulties with the appearance, and often tried to hide the compression hosiery. They mentioned problems with donning and doffing, which can result in dependency of home or informal care, which stopped some participants from wearing. In general, participants were not aware of the full range of assistive products for donning and doffing, but were interested in them. In conclusion, wearing compression hosiery has a large impact on a person's life because of its lack of comfort, unattractive appearance, and possible loss of independence through the need of donning and doffing support. These are expressed reasons for non-adherence, in addition to a lack of understanding of the importance of wearing and the consequences of not wearing compression hosiery. Easy-to-find independent information and more attention to donning and doffing during the fitting appointment of compression hosiery are recommended.

**Data Availability Statement:** All 19 transcript files are available from the DataVerseNL database https://dataverse.nl/dataset.xhtml?persistentId= doi:10.34894/B4UDSO.

**Funding:** This qualitative study was conducted in preparation of a cost-effectiveness study on an optimized provision process of assistive products for donning and doffing compression hosiery. The cost-effectiveness study is funded by ZonMw, project number 10310042110001. "ZonMw programs and funds research and innovation in health, healthcare and well-being, encourages the use of this knowledge and highlights knowledge needs." (https://www.zonmw.nl/en) The grant was awarded to the organizations involved, rather than to individual researchers. The funders had no role in study design, data collection and analysis, decision to publish, or preparation of the manuscript.

**Competing interests:** The authors have declared that no competing interests exist.

## Introduction

Medical compression hosiery (CH) is a treatment option for venous and lymphatic diseases such as varicose veins, venous insufficiency, venous ulcers, lymphedema, lipedema, and acute deep venous thrombosis [1] and lipedema [2]. The prevalence of CH wearers varies from one country to another. Two percent of the Dutch adult population wears CH [3]. In 2023, CH was in the top 3 most provided assistive products in the Netherlands that are reimbursed by health-care insurance [3].

Despite the proven benefits and enhanced quality of life associated with wearing CH [4], there remains a high non-adherence rate of 34% [5]. Non-adherence can exacerbate disease progression and, consequently, lead to reduced mobility, pain, poor quality of life, and increased healthcare costs [6]. Users' experiences of wearing CH have been reported in several studies, mainly with a focus on non-adherence. They describe various reasons for non-adherence including a lack of knowledge, discomfort (pain, heat, and skin irritation), and problems with the appearance of CH [6–10]. Moreover, donning and doffing CH are burdensome and tedious tasks. Twenty-seven percent of CH wearers are not able to don CH independently [7]. This inability leads to non-adherence or dependence on informal carers or home care and, consequently, additional healthcare costs due to complications. About one third of Dutch CH wearers receive home care for donning and doffing. Of note, in the Netherlands home care is already stretched by understaffing. Assistive products for donning and doffing CH (APD) are available to simplify donning and doffing. A commonly used term for these devices has not been identified in the existing literature. For the purposes of this study, the term "APD" is used for both the singular and plural forms. With an APD, 93% of users can successfully don their CH [7]. Nevertheless, only 15% of Dutch CH wearers receive an APD from their health insurance [3].

The available studies on CH provide a relatively superficial description of the experiences of CH wearers and do not provide insight into their experiences using APD. A deeper understanding of the experiences and needs of CH wearers can be beneficial to optimize the provision of information about CH and APD to enhance adherence and independence. Thus, we aimed to describe users' experiences and needs related to obtaining, wearing, donning, and doffing CH and obtaining and using APD.

### Textboxes

**Textbox 1.** Compression hosiery (CH) is manufactured in either a flat or circular knit by using elastic yarn. CH is available in standard sizes or can be custom made, both in various pressure classes I-IV (I: 14–21 mmHg, II: 23–32 mmHg, III: 34–46 mmHg, IV: >46 mmHg) and different lengths (below the knee, thigh length, and panty hose) [2].

**Textbox 2.** In the Netherlands, healthcare insurance reimburses two pairs of CH per year. A physician, mostly a general practitioner, dermatologist, vascular surgeon, or internist, makes the diagnosis. In most cases, the physician indicates what type of CH should be used. The provision of CH is often preceded by bandaging in case of edema. Patients are free to choose the compression therapist (CT) who fits their CH. For non-ambulatory people, CH is fitted at home. In general, regular sizes or ready-to-wear CH

are held in stock and patients receive them immediately; otherwise, CH has to be ordered with a maximum delivery time of 3 weeks. Some CTs first provide one pair of CH and arrange a check-up after 6 weeks. Others provide two pairs immediately and there is no further check-up.

**Textbox 3.** Assistive products for donning and doffing CH (APD) provide support when donning and/or doffing CH is difficult, due to limitations in bending, reduced hand function, or CH with a high compression (> 32 mmHg). They are available for CH wearers as well as (informal) caregivers. Some APD are only suitable for donning, others only for doffing, and some for both. There are more than 35 different APD that can be classified into four categories: (1) grip-enhancing aids, (2) resistance-reducing aids, (3) arm lengthening devices, and (4) electrical aids (Fig 1).

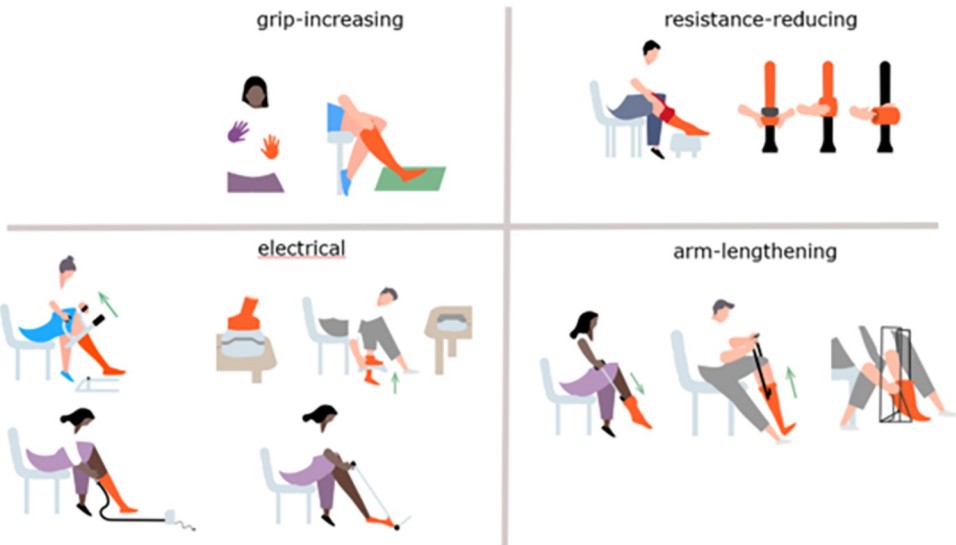

**Fig 1. Categorisation of assistive products for donning and doffing compression hosiery.**

## Materials and methods

This study used a social constructivism research paradigm. This paradigm renders possible to understand and describe human nature with its multiple subjective realities and truths [11]. This descriptive, qualitative study used the lens of CH wearers, focusing on their feelings, perceptions, and experiences [11]. Individual semi-structured interviews were conducted to obtain an in-depth understanding of wearing CH and using APD.

### Study population

The study population consisted of compliant adult wearers of CH with a compression class of II, III or IV (>.22 mmHg) who had been wearing CH for between 6 months and 3 years, so

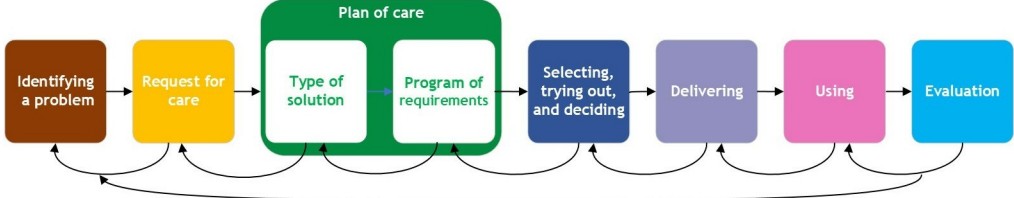

**Fig 2. The Nictiz generic quality framework care for assistive products.**

that each person had sufficient experience and could remember the provisioning process. People were excluded if they were living in a nursing home, had severe cognitive impairments, or did not speak Dutch. Purposive sampling was used to maximize variations in experiences and impacts [12] as well as gender and the type of CH. As a result, more non-compliant CH wearers were interviewed.

## Interview guide

The interview guide (S1 File) comprised six main topics: (1) the CH provision process, (2) the experiences of wearing CH, (3) the experiences of donning and doffing CH and the use of APD, (4) the provision process of APD, (5) the personal characteristics that influence acceptance of assistive products, and (6) the role of the social and physical environment. The topic list was guided by two frameworks [13]. The Nictiz Generic Quality Framework Care for Assistive Products [14] (Fig 2) provided the subtopics regarding the CH and APD provision process. This framework follows the steps for the provision of assistive products; from a person is aware that he has a problem and ends when, during the evaluation, it becomes clear that the person has a properly working assistive product.

Subtopics in personal characteristics and the role of social and physical environment emerged from the model of assistive technology acceptance [15] (Fig 3). In Phase A, potential users of assistive products (AP) encounter difficulties in performing daily tasks and explore the benefits of using an AP, gathering information from others. This stage is characterized by ambivalence and the influence of family attitudes, with a strong emphasis on maintaining independence and being in charge. The transition to Phase B occurs if the task is deemed worth the effort to preserve autonomy. In Phase B, users shift from ambivalence to a belief that AP can improve their daily functioning. They actively seek information about available solutions, with their social network playing a key role. Transition to Phase C happens when the AP are obtained, influenced by trust in healthcare providers and personal perception of the AP' necessity. Once the AP are integrated into daily routines, successful use depends on whether they meet expectations and become part of the user's identity. Encouragement from relatives

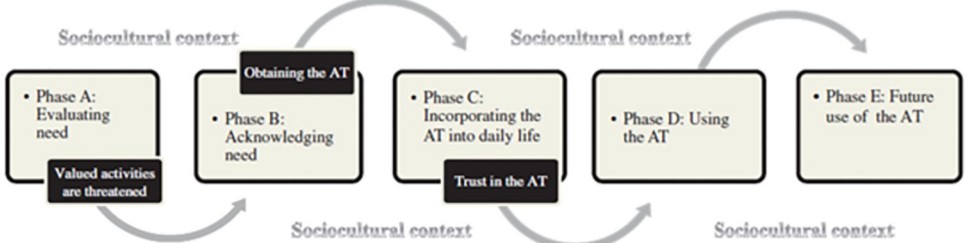

**Fig 3. Phases and transitions in the process of becoming a user of AT.**

also influences the use. Confidence in the AP, especially in their reliability, is crucial for moving into Phase D, where users become competent in their use and experience increased confidence and self-esteem. In Phase E, users consider future use, often hoping the AP will eventually become unnecessary or replaced with a less restrictive option, while some retain the AP as a precaution for potential future needs.

## Procedure

After collecting the demographic information, the interview first focused on the CH provision process and the attention that was paid to donning and doffing during the fitting and delivery of CH. It then zoomed in on the personal experiences of wearing CH, donning and doffing, and using an APD. Lastly, influencing personal characteristics, as well as the role played by the social and physical environment, were examined. All interviews were conducted by the same researcher (EH) who has a Master of Arts degree in change management; she has been trained as an occupational therapist but has mainly been working as a researcher in recent years. She was involved in previous research on donning and doffing CH by caregivers and about usability of APD. A pilot interview revealed that adaptation of the interview guide was not necessary. The data from the pilot interview was so rich and valuable that it was incorporated into the final results. After approval by the Medical Ethical Review Committee METC Z (METCZ20220025), the participants were recruited from 16-5-2022 until 30-09-2022 via patient associations, compression therapists (CTs), home care organizations, and social media. Those who met the inclusion criteria received an information letter about the study. After agreeing to participate, they signed a written informed consent form. Inclusion was continued until saturation was reached. Each interview was conducted face-to-face at the participant's home or via video call, depending on the participant's residence, digital skills, and preference.

## Data analysis

With the participant's permission, each interview was audio recorded and transcribed verbatim. Member checking was utilized: A summary of the interview was sent to each participant within 1 month of the interview to check for agreement. If there was no response within 2 weeks, then agreement was assumed. Interview data were analyzed using directed content analysis, guided by the topics of the interview guide. Coding was conducted by two researchers (EH and UR) using Atlas.ti (www.atlasti.com). First, the interviews were read and re-read to get an overarching view of the content and to get an initial view of the themes and non-pre-conceived topics. EH and UR started by coding two interviews together to calibrate the process. All of the data fit into the coding scheme. After coding all of the interviews, EH and UR ordered similar categories into broader categories and then grouped them into major themes with subthemes. Finally, EH and UR described the key findings by theme [16–18].

## Quality criteria

The following quality criteria described by Frambach et al. [12] were considered.

*Credibility*: The semi-structured interviews were conducted using a predefined topic list, based on a framework for the provision of assistive products. Each participant was asked to provide feedback on the transcript and summary of the interview (member checking). Investigator triangulation was applied as two researchers coded the data.

*Transferability*: Experiences were described in detail and in their context when it mattered to reach thick descriptions and to make the findings meaningful. In the Results, participants are identified by numbers to allow the reader to assess the context. The sampling strategy

strived for maximum variation concerning experiences and impact. In the Discussion, the findings are compared with the existing literature from different settings.

*Dependability*: Data were collected until no new information emerged (saturation). During the interview period, the researcher (EH) was flexible and open to topics and the process (e.g., the inclusion criteria were broadened). By continuously reexamining the data using insights that emerged during analysis (iterative data analysis), it became clear that adding participants who were non-compliant with wearing CH could provide additional interesting data.

*Confirmability*: The interviewer (EH) herself wears CH. She could see the reality through the same "lens" as the participants and thanks to her experience and knowledge, she could ask probing questions to obtain more in-depth information. She reflected critically on her role throughout the research process (reflexivity). Based on the audio recording of the pilot interview, the interviewer (EH) received feedback on her interview technique from a colleague researcher (UR), paying special attention to the neutrality of the questioning (peer debriefing). Steps and decisions taken in the research and their motives were documented to provide an audit trail.

## Results

Each interview lasted for 30–45 minutes. After 19 interviews, data saturation was reached and no new insights emerged. In response to member checking, one summary was modified: The participant felt the summary did not express his experiences appropriately. The summary indicated that the participant did not feel up to learn to don and doff CH himself because of frustration and stress. However, the participant replied that this was not possible because of coordination problems and loss of strength. He added that there was nothing to be ashamed of in asking for help.

### Participants

Table 1 provides detailed information on the 19 participants. The participants wore CH because of various conditions: lymphedema, lipalgia, venous insufficiency, varicose veins, and deep venous thrombosis. One participant could not explain why he wore CH. The participants' backgrounds also varied in terms of the education level and living situation. Fourteen participants wore CH (nearly) every day and four participants wore CH occasionally or never. Thirteen participants could don and/or doff CH themselves, five received help from informal carers, and three received home care. Of those three participants, two were able to don CH by themselves, but they needed help to doff CH. Thirteen participants used an APD.

### Themes

Three main themes were revealed: (1) becoming and being a compression hosiery wearer (or not), (2) wearing compression hosiery, and (3) donning and doffing compression. Within these 3 themes a total of 14 sub-themes were identified. Table 2 lists the retrieved themes and sub-themes.

**Becoming and being a CH wearer (or not).** *Feeling overwhelmed or relieved.* The participants had different reactions regarding the medical advice for wearing CH. Certain participants, such as those dealing with long-standing conditions like lymphoedema or lipoedema and who frequently received a late diagnosis, generally said they took a positive attitude, expecting that CH would bring relief to their ailment (P7 and P11). The participants with acute conditions (P3, P9, and P15), such as deep venous thrombosis (DVT), reported feeling overwhelmed by the news that they would have to wear CH.

**Table 1. Characteristics of the participants.**

| | Age (years) | Gender | Years of wearing CH | Type of CH | Compression class | Who is donning/doffing? | Use of an APD | Family situation | Education level | Diagnosis | Other diseases/disabilities | Adherence |
|---|---|---|---|---|---|---|---|---|---|---|---|---|
| P1 | 57 | Female | 3 | Thigh, circular knit, made-to-measure, open toe | Left: 3 Right: 2 | Self | Silk sock and gloves | Married | Bachelor | Secondary lymphedema | Arm fracture (healed) | + |
| P2 | 29 | Female | 1.5 | Pantyhose, flat knit, made-to-measure | 2 | Self | Only gloves | Single | Bachelor | Lipalgia | - | + |
| P3 | 21 | Female | 1 | Knee only left leg, flat knit, ready to wear, open toe | 2 | Self | Foot slip | Living with parents | Bachelor | Deep venous thrombosis | - | + |
| P4 | 80 | Male | 2.5 | Thigh, flat knit, made-to-measure, open toe | 3 | Self | Foot slip | Living together | Master | Secondary lymphedema | Prostate cancer (cured) | + |
| P5 | 48 | Trans man | 4 | Knee, flat knit, made-to-measure, open toe | 2 | Informal care | Only gloves | Single | Vocational college | Lipalgia and lymphedema | Transient ischemic attack (two times), obesity, prolapses | + |
| P6 | 87 | Female | 2 | Knee, flat knit, flat knit, ready to wear, open toe | 2 | Donning: self Doffing: informal care | Foot slip | Widow | Primary education | Edema | Arthrosis | + |
| P7 | 35 | Female | 1 | Knee, made-to-measure and ready to wear, closed toe | 3 | Self (laborious) | - | Long-distance relationship | Master | Primary lymphedema | - | + |
| P8 | 83 | Male | 2 | Knee, circular knit, ready to wear, open toe | 2 | Home care | Foot slip or Doff N' Donner | Widow | Vocational college | Participant did not know | Partial spinal cord injury | + |
| P9 | 69 | Female | 3 | Knee and thigh, circular knit, ready to wear, open toe | 2 | Home care | Foot slip | Married | Primary education | Deep venous thrombosis | Crushed arm | + |
| P10 | 81 | Female | 4 | Knee, circular knit, ready to wear, open toe | 2 | Home care | Silk sock | Widow | Primary education | Edema | Loss of strength, dizziness | + |
| P11 | 63 | Male | 2 | Knee and thigh, flat knit, made-to-measure | 4 | Self | Silk sock | Married | Vocational college | Secondary lymphedema | Prostate cancer (cured) | + |

(*Continued*)

**Table 1.** (Continued)

| | Age (years) | Gender | Years of wearing CH | Type of CH | Compression class | Who is donning/doffing? | Use of an APD | Family situation | Education level | Diagnosis | Other diseases/disabilities | Adherence |
|---|---|---|---|---|---|---|---|---|---|---|---|---|
| P12 | 69 | Female | 1 | Knee, circular knit, ready to wear | 2 | Informal care | Doff N' Donner | Living together | Vocational college | Edema | Arthrosis, breast cancer | + |
| P13 | 78 | Male | 3 | Knee, circular knit, ready to wear, open toe | 2 | Informal care | Foot slip | Married | Vocational college | Venous insufficiency | Arthrosis in hip | + |
| P14 | 61 | Female | 5/6 | Pantyhose, flat knit, made-to-measure, closed toe | 3 | Donning: self (laborious) Doffing: informal care | Only gloves | Married | Bachelor | Primary lymphedema | - | - |
| P15 | 50 | Female | 6 | Knee, circular knit, ready to wear, open toe | 2 | Self | Foot slip | Married | Master | Deep venous thrombosis | - | + |
| P16 | 53 | Female | 6 | Thigh, circular knit, ready to wear, open toe | 2 | Self | - | Married | Bachelor | Varicose veins | - | - |
| P17 | 71 | Female | >50 | Knee, circular knit, made-to-measure, open toe | 2 | Self (laborious) | Foot slip | Single | Primary education | Varicose veins, edema | Cardiac failure, diabetes, chronic obstructive pulmonary disorder, tendonitis, fibromyalgia, obesity | +/- |
| P18 | 50 | Female | 24 | Thigh, circular knit, ready to wear, open toe | 2 | Self | - | Married | Bachelor | Varicose veins | - | - |
| P19 | 59 | Female | 25 | Knee, circular knit, ready to wear, closed toe | 2 | Self (laborious) | - | Married | Vocational college | Varicose veins | Diabetes, Lyme disease, fibromyalgia, back pain | - |

Before obtaining CH, the participants expressed that they could not imaging the impact of wearing it (P8 and P15). During the CH fitting process, they felt that the impact of wearing CH was often overlooked by both physicians and CTs (P3, P6, P7, P15, and P19). Even when attention was given, the participants said it often had little impact due to the overwhelming amount of new information they were provided. Some participants missed the attention given to the impact of wearing CH; others did not. In addition, CH provision frequently came after bandaging, which had an even larger impact than wearing CH. Hence, the impact of wearing CH was initially perceived to be relatively low to the participants. They indicated that there was a greater impact when they ultimately realized that wearing CH is usually a lifelong commitment.

**Table 2. Revealed themes and subthemes.**

| Themes | Subthemes |
|---|---|
| **Becoming and being a CH wearer (or not)** | Feeling overwhelmed or relieved |
| | Feeling not well-informed |
| | Weighing the pros and cons |
| | Feeling supported by relatives and indifferent with reactions to others |
| | Handling CH with care |
| **Wearing CH** | Experiencing relief and functional gain |
| | Constant dealing with discomfort |
| | Avoiding CH attracts attention |
| **Donning and doffing CH** | Learning to cope with difficulties |
| | Finding the right place in different circumstances |
| | Appreciating independence |
| | Willingness to use an APD depends on the need and perceived benefits |
| | Being unaware of the range of APD |
| | Satisfied about the use of APD |

Many of the participants minded having to wear CH (P1, P2, P3, P7, P12, P14–P16, P18, and P19). Others had little or no problem with wearing CH and were not ashamed of it (P4, P6, P8, P11, P13, and P17)—remarkably, all men and women over 70 years of age. They felt wearing CH is appropriate for their age (P6 and P17).

> P1: *Accepting that you have lymphedema, if you don't have it from birth, and wearing CH becomes much more manageable if you have the right guidance and support. With a dermatologist or skin therapist who listens to you and thinks with you.*

> P1: *That has to do with my ability to put things into perspective and also when your husband goes blind and you get cancer, how bad is this? You can't compare and I also sometimes think of myself as the most pathetic person in the whole world, but that is the way things are and you don your CH and you go on.*

*Feeling not well-informed.* In many cases, the participants felt that they had not been fully informed by physicians, especially regarding the aesthetic aspects of CH (P7 and P15). During CH fitting by their CTs, the participants received extensive information, but they did not remember everything. Short appointments hindered questions and discussions, resulting in unresolved queries during CH use (P9 and P17). The participants explained that during the fitting appointment, they felt confronted with a lot of new experiences (P10 and P17) and felt they had missed a lot of the verbal information they had received. A few felt well informed (P1, P9, and P11); they had stayed in a specialized treatment center for some weeks and had received all the required information. The participants stressed that good advice and support from healthcare professionals helps with the acceptance process (P1, P2, P4, P5, P9, and P11). Sometimes, CH was fitted by a skin (edema)therapist (ST) instead of a CT. A few participants (P2, P4, and P5) indicated that they appreciated the approach of the ST who was knowledgeable, took time for explanation and emotions, answered questions, treated lymphedema, and paid attention to donning and doffing, including fitting an APD.

Most of the participants were unaware of the specific type of CH they were wearing. Almost all of the participants believed they were wearing made-to-measure CH, but only 8 of the 19 participants were actually wearing made-to-measure CH. The majority could point out why

they must wear CH and the implications of not doing so (P1–P5, P7, P9, P12, P14, P16, P18, and P19), but three participants (P6, P8, and P13) expressed a lack of awareness of the reasons and significance behind wearing CH. Two participants (P6 and P10) did not know that their health insurance covers the cost of two pairs of CH each year, leading them to wear the same CH for multiple years.

Most of the participants commented on the limited freedom of choice and shared decision-making regarding CH selection. For example, the choices in CH were primarily restricted to color (black or skin tone), toe type, and, occasionally, length (P1, P3, P6, P8, P12, and P14–P18). Only two participants (P4 and P11) mentioned that factors such as activities and personal circumstances were considered, and they appreciated the fact that decisions were reached through consensus. The participants emphasized the importance of evaluation after a few weeks.

According to the participants, practical information is needed regarding wearing CH, fit (tightness/looseness), wearing duration, suitability in warm weather, maintenance (washing/drying), and CH lifespan (P3, P6, P7, P10, P15, and P17). The participants often found information leaflets about CH to be unreadable due to small text and multiple languages (P3 and P5). Some participants sought insights from other CH wearers and information on recent advancements in compression therapy. The participants who searched the Internet found fragmented and sometimes conflicting information, primarily on commercial websites (P1, P3, P5, P14, and P15). The independent information sources mentioned by the participants comprised websites of patient associations of DVT and lymphedema/lipalgia, and a specialized treatment center for lymphedema. Additionally, some mentioned the patient associations on social media. The participants emphasized the significance of critical evaluation when reading information, as it may not always be impartial or reliable. P3 thought that tight trousers enhance the effectiveness of CH and that CH color influences its effectiveness.

P3: *I found it [information] on the Thrombosis Foundation website [. . .]. Other than that, I've gone from website to website. I went over English and American websites, as well as a couple of Belgian websites and a French one. Then you just read through that and then you look at which things generally come up multiple times. So, I would have liked it if there was just one website.*

P5: *With the stocking, you get such a thick booklet and then I still don't know what to do with it. [. . .] It's not very informative. [. . .] A compact booklet would have been good enough. And then in 20 languages and small print, I'm not going to bother with that.*

*Weighing the pros and cons.* Most of the participants perceived CH as a necessary evil (P1–P13, P15). They consistently wore CH on a daily and continuous basis due to the perceived positive effects or concerns about potential risks associated with discontinuation. However, the participants who reported experiencing no immediate benefits from CH (P14, P16, P18, and P19) tended to wear them inconsistently, despite the associated risks of wounds or infections. One participant (P17) believed that there was no point in wearing CH occasionally, so she did not wear CH at all. Two participants (P14 and P19) selectively wore CH when suffering or as a form of prevention, for example, during extended travel. The reasons the participants mentioned they were not willing to wear CH included pain or discomfort (P14, P16, P18, and P19), the appearance of the CH (P16), and problems with donning and doffing (P14 and P17). On warm days, some participants explained that when wearing CH, they experienced more symptoms of the disease (P17), whereas others abstained from wearing CH because the heat makes it uncomfortable (P16 and P19). P1, P7, and P14 augmented CH with a pneumatic

compression device. Due to discomfort, some participants explored alternatives such as padding bandages (P14), over-the-counter compression class 1 (< 22mmHg) CH (P18 and P19), or tightly fitted skinny jeans (P14). Some participants expressed that the effort required for donning or doffing CH prevents them from wearing it daily (P2, P17, and P19). CH that is easier to don (e.g., with a zipper) would increase willingness to wear it (P17). P5, P14, P15, and P17 struggled with improperly fitting CH: It was too long or too tight, sometimes leading to inconsistent usage.

> P11: *If I didn't have to wear them, I wouldn't wear them. Nobody does it for fun. It's the same with hearing aids or glasses, if you don't need them, you don't put them on.*

> P19: *It's not that I don't want to wear them, is it? It's just that there's so much stopping me from wearing them.*

> P18 [woman with varicose veins and occasional phlebitis]: *I think the discomfort of wearing outweighs the occasional symptoms.*

*Feeling supported by relatives and indifferent to reactions of others.* The participants clearly stated that they decide whether or not to wear CH. The role of relatives was mentioned by several participants (P2, P4, P5, P10, P11, P12, P14, P15, P16, and P18). The participants stated that family members sometimes encouraged wearing CH (P17 and P19) or were understanding and willing to support the participant when necessary—for example, in donning (P1, P5, P6, P12, P13, P14, and P15). Some relatives helped to find solutions to problems regarding CH. For example, P4's partner modified his CH so that it no longer sagged. Another repaired CH with a hole (P9).

The participants mentioned that negative reactions most often come from people who are less close to them, for example, people telling them how horrible it must be to wear CH (P11). Most of the participants said that these reactions have had no impact as they wear CH because it is necessary, regardless of others' opinions. They usually tended not to invest any energy in trying to change people's minds. The participants explained that they usually do not talk to others about wearing CH, nor are they looking for peer contact to share experiences (P1, P2, P4, P7, P9, P13, P14, P15, and P17). A few were members of a patient association or followed its social media (P1 and P3).

> P14: *My husband says: "You are old and wise enough to feel for yourself what you do and what is possible."*

> P17: *My son frequently comments about not wearing the CH regularly at the beginning. "Mom, that's not good." And my daughter-in-law honestly says, "Mom, it's not wise. You should wear them, but I understand why you aren't wearing them."*

> P11: *I also speak to people who say, "I don't understand why you wear those stockings in summer then the leg [is] a bit thicker." Then I say, "That's because you don't know what you're talking about because you don't have this disease. Otherwise you would be wearing them."*

*Handling CH with care.* he participants said they handle their CH with care: They wash them separately (P10) and keep their fingernails short or use anti-slip gloves, as long nails can damage CH (P1). Nevertheless, some of the participants (P1, P6) reported that the stretch of the CH decreases over time, and CH can wear out and become perforated (P2, P7, P9, and P10). P3 (incorrectly) thought that machine washing and spinning should be avoided to preserve CH quality. Two participants (P7 and P9) were satisfied about the warranty. In the

Netherlands, health insurance reimburses two pairs of CH per year. Two participants (P2 and P12) felt that this was not enough because of wear and tear and because they had to rotate CH for washing. The participants found CH to be expensive, keeping them from buying an extra pair (P5). Thus, despite reduced pressure, CH are worn longer than 1 year so that they have more CH to rotate and have more time for washing and drying.

> P2: *That's just wear and tear, so two CH in a year is not enough. I have two with a hole in them now, you wear them out pretty fast because you have to wear and wash them daily.*

**Health benefits yet discomfort from wearing CH.** *Experiencing relief and functional gain*. The participants described that they experienced pressure from CH around their legs. While some found the support beneficial (P6 and P15), others reported it to be unpleasant (P7, P12, P14, P16, P17, and P18). Most participants reported positive effects associated with wearing CH: ankles/legs there were not as thick (P1, P6, P7, P11, P12, and P19), no wounds (P12), fewer varicose veins, no more pain in calves (P2), and enhanced standing and walking (P2). Notably, the participants reported that when CH is not worn, leg swelling occurs (P1 and P7), and some participants experienced discomfort, like problems with walking the following day (P5) or a persistent sense of fatigue (P19). Those who observed positive effects generally considered the benefits to outweigh the drawbacks. However, over time, CH elasticity diminished, and symptoms reappeared, prompting some participants to desire more frequent replacement (i.e., more than once a year) (P5 and P12).

> P2: *But I did notice very clearly that when I was wearing them (CH) I could stand longer at a bar table at a party or walk better than when I wasn't wearing them. By the end of the evening, I'm walking like a penguin. I was waddling, and when I'm wearing these CH I can hold on much longer.*
>
> P5: *If I don't wear them, I can hardly walk the next day.*
>
> P6: *I like the fit around the legs and the support around the legs. I have never taken them off. That's how comfortable I find them.*

*Constant dealing with discomfort*. Wearing CH could cause discomfort, pinching, and even pain according to many participants (P2, P4, P8, P14, P15, and P18), which could also interfere with their activities (P1). A few participants said they could not tolerate wearing CH all day for this reason (P14 and P19). The participants stated that warm weather exacerbated the discomfort, as it tended to make them feel more bothered than during the winter months (P1, P3, P4, P6, P9, P10, P11, P13, P16, P18, and P19). According to some participants, the brand (P13) and length of CH seemed to have an impact on comfort: All wearers of thigh-high CH complained about discomfort. The knitting method also impacted comfort: P2 and P7 tried both and rated the comfort of the flat knit higher. The label inside the CH can cause irritation (P10), but removal will void the warranty (P5). P9 indicated that a skirt or thin trousers can creep up along CH. The majority of wearers of thigh-high CH reported that this type tends to sag (P1, P4, P11, P12, P16, and P18), which could be especially problematic when outdoors, as it requires pulling trousers down or lifting skirts to put CH on properly again (P1 and P4). To prevent sagging, the participants sometimes used skin glue (P11) or opted for tights or knee-high stockings rather thigh-high stockings. P4 sought creative solutions to improve the comfort of CH and to prevent sagging; however, tights could make standing urination impossible for men. P2 mentioned that tights also could cause uncomfortable pressure on the abdomen for women, particularly during menstruation, leading to avoid wearing it. Additionally, the

participants indicated that tights were even warmer than CH and considered them to be too hot in the summer.

> P19: *In the summer it bothers me. When it's so hot. I can't stand the heat very well anyway. So, when I hear that it's going to be a hot day, I don't put them on.*

> P17: *In the summer, when it's so hot, I wear the CH every day. But [. . .] when it's a little less hot [. . .] I'm not wearing them. I have a lot of trouble donning them.*

*Avoiding CH attracts attention.* When the participants discussed the impact of wearing CH, their focus frequently revolved around the visual aspect of it (P6 and P16), and the implications for their clothing and footwear choices (P1, P2, P3, P5, P7, P9, P10, P12, P15, P16, P18, and P19). The participants found their CH to be ugly (P16, P18, and P19) and associated CH with old people (P3 and P5). Many participants expressed issues with the color and material of their CH (P2, P14, and P15). They stated that the color was not a suitable match for their skin tone (P2) and faded with washing. According to several participants, there was little innovation in CH: The look of CH has been the same for decades (P2 and P14). Some suggestions for improvement included using a silkier material, adding glitter to the seam, or incorporating prints that resemble tattoos (P14 and P15). The participants were often unaware that CH are available in hues other than skin tone and black. Other participants felt conflicted about CH colors. They desired more variety to match their clothing but often compromised for neutral colors due to the annual restriction of two reimbursed pairs. P7 opted for made-to-measure CH because of the broader color selection. Some participants said they drew attention away from CH by wearing conspicuous shoes or by wearing colorful CH that are not recognized as such (P3 and P5); others would hide CH under long trousers or dresses (P1, P2, P3, P9, and P10). P1 even bought a new wardrobe to hide her CH. Remarkably, the men interviewed did not report issues about the appearance of CH at all. Some participants reported that wearing CH reduced their positive self-image and affected their intimate relationships (P2 and P18). One participant described it as a process of grieving, of trying to get used to a new situation. It decreased quality of life according to P1. P2 stated she had received psychological counseling to accept the disease and its visible effects, including CH. In contrast, others downplayed the impact of CH concerning a serious condition such as cancer that underlies the wearing of CH (P1, P4, P8, P10, P11, and P12).

> P2: *And what I also find very difficult, I am still single and you still want to meet someone special and you don't feel sexy in such a CH.*

> P7: *Yes, it's annoying. It doesn't combine so nicely. You occasionally see people looking at it like*: *"What is she wearing?" [. . .] Last Friday I had dinner and I arrived by bike in a denim dress with pink CH underneath and they said*: *"Oh, you look nice today, those pink stockings are hip." Then I said yes they are compression stockings. "Oh really?" [. . .] I am not ashamed.*

> P2: *All the skin-colored ones I do find a pity, [. . .] it still does not match well with your normal skin. I still think they stand out a lot.*

**Donning and doffing CH.** *Learning to cope with difficulties.* Many participants had additional conditions (Table 1) that limited their mobility and strength and therefore their ability to don and doff CH. Thirteen participants could don or doff CH by themselves, five received help from an informal caregiver for donning and/or doffing, and four received home care. Four participants did not use an APD, six used gloves and/or a silk foot, eight used a foot slip

and two caregivers use the Doff N' Donner. The participants had varying opinions on the ease of donning and doffing CH. While some found it relatively easy (P1, P4, P13, and P16) and less challenging than expected (P7), others reported that donning can be an arduous task (P2, P3, P5, P10, P14, P17, and P19) that even caused physical complaints (P2, and P17). Donning and doffing require dexterity and proper instruction. The participants reported that when their CH was new, it was very tight. Gradually, it became less tight, making it easier to don and doff (P15). The CH type also matters. According to P2, flat-knit CH is easier to don than round-knit CH, and made-to-measure CH is perceived as tighter than ready-to-wear CH. The participants stated that taking one's time during donning and doffing is essential; it is more difficult when in a hurry (P2). The participants also mentioned that if the activity was not carried out smoothly, it could cause temporary localized pressure in one area, which could be painful (P12 and P15). Wrinkles were smoothed with anti-slip gloves during the day by some participants (P4, P12, P15, and P18). The participants were conflicted about which activity was the most difficult. According to several of them (P1, P2, P3, P4, and P19), doffing required less effort than donning. However, others found doffing more difficult and exhausting (P6, P7, P10, P14, P15, and P17).

Some of the participants needed help at first, but after habituation (P3) or recovery (P15), they could perform independently. The method of donning and doffing changed over time. The participants established their routine and learned tricks, which were not always recommended or optimal for their CH (P2, P4, P5, P7, and P15). For example, P4 used talcum powder to make his legs smoother, and P5 opted for shorter CH instead of tights due to difficulty removing them in time to use the bathroom. Some participants chose less snug (older) CH if they needed to doff and don them multiple times a day, as the effort involved in donning tighter CH was deemed too burdensome. Two participants (P1, P10) reported that physical conditions and social circumstances, such as a surgery or the death of an informal caregiver, may result in temporary or permanent changes that rendered donning and doffing infeasible.

The participants indicated that donning and doffing CH affected their self-care routine, requiring more time and prompting reevaluation of morning rituals, such as showering times. Over time, new routines became habitual (P13).

> P17: *But that donning and doffing. . . It's just a disaster. You should see me here alone in the evening. And, then I have to put myself in all kinds of twists and turns. I pulled a muscle in my but and I can't do anything.*

*Finding the right place in different circumstances.* The participants experienced that for donning and doffing, it was important to have a good, stable sitting position, enough space, and a table or cabinet to put all the necessary items. Most people donned and doffed their CH in a fixed place, often in the bedroom or bathroom, sitting on the toilet. The participants mentioned that it was very difficult to don CH under certain circumstances, such as after showering or swimming or donning a tight in a portable toilet at a festival (P2, P4, and P7).

> I: *Does it matter to you where you don the CH at home?* P2: *I like to be able to sit, so I either sit on the toilet or the couch. That way I have space to stretch my legs properly and then I sit a little lower so I can bend a little easier. I can also do it standing up, but then it takes more strength, sitting down is more comfortable.*

*Appreciating independence.* The participants noted that when they were fitted for CH, some attention was paid to donning and doffing. However, this attention was often brief and more focused on donning than doffing. If independent donning and doffing was possible, then there

was a narrow focus on optimization (P2–P5, P7, P9–P11, P13, and P15). In several cases, there was no attention given at all to donning and doffing or the use of APD (P1, P6, P12, P14, P17, and P18). None of the participants contacted the CT afterwards if donning or doffing was still difficult or became difficult over time. Some participants were unaware that they could seek support and training from an occupational therapist (OT) to learn how to don and doff CH independently (P12, P17, and P19). An OT was only occasionally involved in the home care service to achieve independence (P5 and P9). Sometimes an OT was involved but the participant did not know that this carer was an OT (P9).

Independence was important for most participants (P1, P2, P6, P8, P11, P12, P13, P14, P15, P17, and P19). The participants indicated that being dependent on home care means losing your freedom (P6, P9, and P10): Waiting on home care limited their ability to plan the day and to participate in activities outside the home. P2 would even consider surgery if independent donning and doffing was no longer possible. When a partner assisted with donning, the participants and their partners became attuned to each other. Sometimes they did it together (P13). Partners did not see it as a burden (P9, P12, P13). However, not all participants' partners could provide informal care due to their own health issues or practical limitations (P9).

Three participants received home care for donning and doffing CH (P8–P10), and one participant received home care only when her husband had a morning shift (P12). Some participants had already received home care before they received CH because of leg bandages or for assistance with other care activities such as wound care or showering. Sometimes, temporary home care was used (P10) to don and doff CH when the participant was less functional for a while, such as after a fall or surgery. The participants felt that home care workers did not always encourage their clients or informal carers to don and doff CH themselves. Some participants experienced professionals who told them it was not possible to do it independently and took over the care (P8 and P9). According to P5, P9, and P10, home care workers sometimes practiced the use of APD with them to promote self-sufficiency. None of the participants recalled that the full range of APD was presented to them even if self-reliance was not achieved. According to the participants, most home care workers used a foot slip or a Doff N' Donner to relieve their workload. The participants believed that not all home care workers were skilled at using an APD. In a few cases, the family carer had to explain to the home care worker how to use the APD (P12). According to P8, some home care workers had a personal preference for a particular APD and then took it from client to client. Home care was also provided during holidays, and even abroad (P9).

I: *Do you wear the CH the whole day?* P5: *Yeah, from about 7 a.m.* I: *Right when you get up*? P5: *No, not right when I get up because I'm already up at 5:30 and then there is no help available. So between 7 and 8 in the morning I receive help to don the CH. I usually wear it until 9 in the evening.*

P10: *Yes, of course I want to do that myself because that's my independence. I mean, I want to stay independent for as long as possible.*

P14: *Yeah, I can do that, but it's a killer job. I do early shifts and I start at 4:30 in the* P9: *The women [home care] who don them also said*: *"It's a hopeless task, you can hardly get them on yourself."*

P17: *There was a time they [home care] came to don. And then I said*: […] *"I can't sit with my legs up until 11:30. I'm alone, I have to prepare my food. Because of my diabetes, I have to eat regularly."* […] *Well, that [waiting on home care] doesn't work for me.*

*morning. So I get up at 3:30 in the morning and if I want to don the stockings I have to get up at least 45 minutes earlier to do this. No home care comes in to help at that time. So, I have to don the CH myself. I have these nice rubber gloves that allow me to don them properly.*

*Willingness to use an APD depends on the need and perceived benefits.* Many participants had a positive view of APD and were open to using one if it were effective (P4, P7, P8, P9, P10, and P17); with the exception of P7, they were using an APD at that time. Some participants stated that they would only start using an APD if donning and doffing were to become very difficult or impossible (P1, P2, P6, P7, and P10); they felt they were more restricted when using an APD. The mentioned several reasons for using an APD, including maintaining independence (P10, P13 P15, and P17) and making donning and doffing easier (P4 and P16). The participants indicated that an APD should provide a clear benefit and be easy to use (P2, P7, and P9) and it should not cause frustration (P5 and P9). If the use of an APD caused pain, one participant preferred to accept help (P9).

Most participants who had an APD said they used it every day (P4, P12, P13, and P15). APD were used because they made donning and doffing CH quicker and easier (P3, P4, P11, P12, and P16), and to prevent damage to the hand (calluses/blisters) (P2). A few participants did not use their APD daily but chose to when it added value—for example, after showering when legs were still clammy and donning was more difficult (P3). The reasons for not using an APD included no need because donning and doffing could be done without an APD (P1, P2, P7, and P11), using an APD caused stress (P5), being unaware which APD existed (P17 and P19), the price of APD (P10), and independent donning/doffing was not possible with the use of an APD (P5, P8 P9, P10).

P2: *Yeah, if it can save my back a little bit I would definitely like to try it.* I: *Would you also go down the route of getting advice on it?* P2: *Yeah, I think so. If it would help me in the future.*

P17: *But I'm a little annoyed. That I suddenly hear, thanks to you, that there are people who can help you. Who could have the tip of the century? And that all these years, I'm 71, I've had these shitty things in my closet since I was 12, nobody ever told me that. That's a shame.*

*Being unaware of the range of APD.* According to several participants (P3, P4, P13, P15, and P16), they were provided with a foot slip without consultation, although they did not expect a consultation. In general, the participants were aware of the existence of APD but were not familiar with the whole range of products. Foot slips and frames were mentioned. Non-slip gloves were not recognized as an APD (P1), and arm extending and electric APD were unknown. P17 was completely unaware of the existence of APD. The participants said that in general, they received little information about APD from CTs or home care workers, nor did they look for information themselves. Sometimes, the participants tried several APD before making a choice (P5, P9, and P10).

Once the interviewer gave information about the types of APD, several participants expressed great interest and asked for more information (P2, P6, P7, P14, P17, and P19). Several participants intended to contact their CT or an OT as a result of this information. P17 emailed a few weeks after the interview to report that she had been seen by an OT and was now able to don and doff her CH independently using Handylegs and a foot slip. As a result, she said show now wears her CH every day instead of occasionally. The participants were often unaware of the cost of an APD and how it is reimbursed (P4, P6, P11, and P15). A few ordered and paid for an APD themselves (P2, P6, and P10), even though it is covered by health insurance. P10 had heard about a new electric APD but did not want to request it because of the cost.

P4: *Yeah, I don't know, are there other APD? Yes, I am very interested. Are they better than this one, easier to use than this one?*

P10: *They have a new APD now. I heard about it from [name of home care organization]. It's an electric APD to doff the CH, but it's very expensive. Then I have to go to an occupational therapist. I'm thinking, "I'm already so expensive for the health insurance company....!"* I: *Well, what do you think home care costs?* P10: *Really?* I: *That's more expensive in the end than the electric APD. Home care costs money every day.* P10: *Every day? Every day?!* I: *Yes, in the end, the electric APD or whatever device you use is much cheaper than home care every day.* P10: *I didn't realize that.*

*Satisfied with the use of APD.* Half of the participants used an APD, mostly a foot slip (for details, see Table 1). The "silk sock" that comes standard with a CH with an open toe was considered less effective than a foot slip by P4, while P1 and P11 preferred this silk sock to a foot slip. The Doff N' Donner was used by the (informal) carer and not by the participant (P8 and P12).

The participants were generally satisfied with the functioning of the APD, mostly a foot slip (P3, P4, P9, P12, P13, and P15). However, some participants had difficulty pulling the foot slip out of the open toe when donning (P4 and P15). The participants did not know of tricks to ease this difficulty. Opinions differed on the durability of a foot slip. P15 was satisfied with the durability whereas P6 and P13 found that it wore out quickly and last only 1 year. P12 thought it is important that an APD fits into a suitcase so that she could take it on holiday.

According to P12 and P15, most APD require practice to become proficient, but not all of the participants held the opinion that they had received good instruction (P3, P12, P13, P14, P15, and P17). It was only used once or a few times under supervision during CH collection. After that, the participants had to be able to do it on their own. For P6, the APD was delivered by post without instruction. P12 received the APD from home care, allowing her informal carer to take over, making home care largely unnecessary. In some cases, the use of an APD was not successful at all or caused stress and independence was not feasible (P5 and P9). The participants did not mention anything about evaluation of APD.

P5: *I've had several APD. One thing you had to slide it over and then you rolled it back up and then you rolled it again. That was so many actions and with that. I didn't get that. [...] There was another water thing, a kind of water bag with it. I couldn't get it to go from the head (brain) to the hands. I thought, "forget it." [...] I also tried a metal frame, but then I had to slide the CH over it, but I couldn't. I didn't have the strength.*

P15: *At the beginning, everything was much more difficult and painful. Because the leg was still very swollen and I couldn't bend the leg properly [...]. But I also lost a few pounds, so I think that played a part as well. I was also stiffer, I don't know. And the dexterity and the instructions that I've gotten since then.*

## Discussion

In this study, we aimed to delineate the experiences and needs of CH users related to wearing, donning, and doffing CH, and the provision process of CH and associated APD. We identified several subthemes that capture the experiences of becoming and being a CH wearer (or not), wearing CH, and donning and doffing CH. This study has revealed the variance of reactions to wearing CH: The prescription of CH can overwhelm people if there is an acute problem or give them hope if they have been experiencing symptoms for a long time. These differences in

responses have not been reported in other studies, which might be a result of a focus on target groups with a single disease in other studies. Based on the findings, becoming a CH wearer has a marked impact because of the negative image of CH (ugly and associated with older age). Finlayson [319] reported in her study that wearers of CH felt associated with disabled people. The finding that CH wearers often feel poorly informed about wearing and maintaining CH has been reported previously [4,10,20,21]. However, this study has added insight into the consequences: For many CH wearers, feeling that they were not well informed caused feelings of insecurity and could result in non-adherence. This is in line with the findings of Finlayson [19]. Although this appears to be obvious and the situation could be easily improved, it is remarkable that the provision of information is still perceived as inadequate.

Most of the participants viewed being a CH wearer to be challenging. Issues with discomfort and the appearance of CH causing non-adherence have been presented in other studies [6,8–10]. In addition to previous studies, we collected the positive and negative effects of wearing CH and explored the impact of these effects on daily life and psychological well-being. For example, the tendency of CH to sag is well known, but the effect of having to lift CH when away from home has not been described previously. This perspective broadens our understanding of the reasons users choose not to wear CH and provide points to address to ensure clients are better informed.

We found several ways users dealt with the discomfort of wearing CH, including wearing CH only if edema increased or as a preventive measure when edema is expected to increase due to weather or activity. These strategies have also been reported by Gong et al. [6] and Probst et al. [10].

Despite the availability of APD, the participants in this study shared problems with donning and doffing CH, a finding similar to previous studies [7,10,20–22]. Some participants accepted help from informal carers or home care because they felt there was no alternative. However, not everyone accepts the loss of independence and, therefore, they may decide not to wear CH, even though not wearing CH has physical risks. Probst et al. [10] and Chitambira [21] also described that the acceptance of professional support and the resulting loss of autonomy can be a barrier to wearing CH. We found that our participants were not well informed regarding many APD; they reported not being informed about APD during the CH fitting process. Previous studies have also shown that CTs and OTs are not fully aware of the range of APD and their usability [23]. The lack of knowledge about APD may be one of the reasons why the potential of APD is not fully exploited and only a relatively small proportion (15%) of those who wear CH use an APD [3]. More attention to the range of APD and their applicability in the vocational training and continuing education of CTs, OTs, and district nurses may have a positive impact on the independence of CH wearers.

Due to its classification as a medical device, individuals in need of CH transition from being consumers with free choice to patients with limited choice: They need care and reimbursement from health insurance companies [24]. Some participants in this study described the process of becoming a CH wearer as one of grieving and trying to get used to a new situation. In contrast, others downplayed the impact of CH regarding their serious condition. Our findings indicate that many factors impact the acceptance of CH. Next to the physical effect of wearing CH, the appearance of CH and the effect on self-image frequently plays a major role. Vreeswijk [24] noted that the relationship between the appearance of the assistive product, the cultural context, and current fashion influence the meaning users attribute to an assistive product. CH are visible and part of clothing, but they can hardly be adapted to one's fashion preferences. Just as Vreeswijk [24] described, our participants stated that they felt unattractive, embarrassed, and old wearing CH, and they placed value on the choice of color of the CH. Vreeswijk [24] explained that the appearance of CH is important in coping with a new

situation and disciplining oneself to wear CH every day. So, the CH color choice is not simply a "luxury"; rather, it is a factor that influences effectiveness because it plays a role in acceptance and participation in daily life situations [24].

Most participants said they wear CH consistently due to the benefits they experience. For some, motivation lay in the fear of the consequences of not wearing CH, such as wounds or recurrent thrombosis or varicose veins. These findings are consistent with previous studies [6,10,21]. The participants who did not comply stated the following reasons for their non-adherence: lack of short-term benefits, appearance, comfort, and inability to don and doff CH independently. Finlayson [19] reported high rates of non-adherence (47%) largely explained by difficulties with donning or doffing, misconceptions about function and wearing duration, and problems related to the unaesthetic appearance of CH. The participants expressed their willingness to use an APD if it were to provide a clear benefit. If not, then the convenience of using an APD does not outweigh the impact of using it, such as feeling more restricted and experiencing stress. The listed factors for accepting the use of both CH and APD correspond to the determinants—perceived usefulness, ease of use, and benefits—that have been identified as success factors for the adoption of a product in several specific models such as the technology acceptance model [25] and the unified theory of acceptance and use of technology [26]. Other models like the diffusion of innovation model [27] and the model of pre-implementation acceptance [28] also emphasize the influence of the social environment on the decision to use a product. Our study also revealed how the social environment impacted the use of CH and APD.

Based on our findings, more attention should be paid to the many factors that contribute to the process of becoming a CH wearer. To increase adherence, caregivers should not only support the use of products, but also focus on acceptance of the new self-image. Mayer-Larssen et al. [15] recommended a client-centered approach in the provision of assistive products. The provision process might benefit from the model of Bloem and Stalpers [29], which emphasizes the relevance of assessing a client's motivation and readiness and collaboratively setting goals and action plans.

The participants remembered having received a lot of verbal information during the CH fitting appointment, but most were unable to recall exactly what was said. Some participants held incorrect beliefs such as tight trousers enhance the effectiveness of CH, machine washing and spinning should be avoided, and CH color influences its effectiveness. The information leaflet attached to the CH did not meet the needs of the CH wearers. Berszakiewicz et al. [4], Probst et al. [10], and Perry et al. [20] have endorsed the importance of educational interventions with a focus on patient perceptions and understanding. Berszakiewicz et al. [4] and Dale and Gibson [30] have even related non-adherence to a lack of understanding of how CH works. We found that participants have different needs regarding information about CH and APD, and that adherence to wearing CH is about changing behavior, acceptance, and dealing with negative effects of CH. According to Bloem and Stalpers [29], changing habits requires coaching; moreover, the delivery of information should not be standard, but rather based on the acceptance of the chronic illness and the client's perceived control.

Based on our study, some alterations in the provision process of CH and APD might be beneficial to increase adherence to wearing CH and independence in donning and doffing. Users might benefit from objective, non-commercial, easy-to-read information about the functioning of CH, donning and doffing CH, and maintenance. Both digital and non-digital information are necessary, as many CH wearers are elderly and not digitally proficient. More emphasis is needed on the acceptance and impact of wearing CH. Although not fully aware of the range of possibilities, the participants were very interested in APD to maintain their independence. A previous study identified a knowledge gap among healthcare professionals about

APD and bottlenecks in the reimbursement structure of CH and APD [23]. CT education could be extended to include the psychological aspects of wearing CH, as well as training about APD. However, adequate reimbursement of the time spent for educating and training CH wearers and adequate reimbursement of APD is necessary to motivate CTs to apply the new strategies in the provisioning process. As healthcare costs continue to rise and staffing becomes problematic, it would be very useful to study whether changes in the provision process could increase adherence to CH and independence in donning and doffing.

## Strengths and limitations

In contrast to most other studies, we centered our research on people who wear CH regardless of the underlying medical condition. A benefit of the diversity in medical conditions is that we collected wide of experiences wearing CH and the impact of CH. A major strength is that we also included participants who did not (fully) adhere to wearing CH. These interviews provided interesting information, particularly about the reasons for non-adherence. The disadvantage was that the non-adherent participants had often been prescribed CH more than 3 years ago, which may have affected their memory of the provision process, possibly causing response bias.

Due to the wide variation in gender, condition, type of CH, and adherence, we captured a wide range of experiences. Admittedly, the sample is not representative of the general Dutch population that uses CH. The average age and the representation of people with venous disease were lower in our sample compared with the overall population. Further, we only interviewed white native Dutch people. Although we tried, we were unsuccessful in including participants with other ethnic backgrounds in the study. Evaluating the representativeness of the participants proved to be challenging due to the absence of dependable Dutch data regarding the occurrence of CH across various ethnicities.

The fact that the interviewer is a CH wearer herself and thereby an expert by experience brought challenges in addition to the benefits already described, such as being able to empathize and recognize experiences and thus ask probing questions to get deeper insights. Particular vigilance was required to remain objective and not to ask suggestive questions. Nevertheless, the advantages of this situation outweighed the disadvantages. Personal reflexivity was applied throughout the entire study by explicitly disclosing the researcher's prior experiences and motivations [31].

The interviews were conducted in 2022 shortly after the coronavirus disease 2019 (COVID-19) pandemic restrictions in the Netherlands had been lifted. Some participants received a CH for the first time during this pandemic. Due to the restrictive measures, the provisioning process may have been different from normal. For example, in some cases the partner was not allowed to be present during appointments with the physician or CH fitting.

## Conclusion

We found that wearing CH has a large impact on a person's life due to comfort, appearance, and possible loss of independence. Poor comfort—such as sagging, stretch marks, and heat—is a major reason for not wearing CH. The effect of the appearance of CH on self-image is an important but underestimated factor regarding the acceptance of CH. The participants reported feeling unattractive, embarrassed, and old wearing CH. CH are visible and part of clothing, but can hardly be adapted to one's fashion preferences. The participants placed value on the CH color choice. To increase adherence, we recommend insurers to reimburse more than two pairs of CH a year so that the wearer can obtain various colours of CH to adapt them to their clothing. Manufacturers are recommended to improve CH concerning comfort and

appearance e.g. better matching skin tints and a more silky-smooth material especially for measure to made CH.

The reason why wearing CH is important is not always clear, and CH wearers often feel not well informed about wearing and washing instructions. There is a need for independent information. It should be considered that participants have different needs regarding information based on their acceptance of the chronic illness and desired level of self-management, so information and guidance should be tailored.

Independence in donning and doffing CH is important for clients, and loss of independence is a major reason for non-adherence. The use of APD can prevent dependence, but, in general, the full range of APD have not been exploited. More attention to donning and doffing CH and APD during the fitting appointment of CH is recommended. In addition, there should be easy-to-find and accessible objective, non-commercial information on the possibilities of APD.

## Supporting information

**S1 File. Interview guide.**
(PDF)

## Acknowledgments

We would like to acknowledge several people:

All of the participants for candidly sharing their experiences;

Lymfoedeem.nl, Harteraad, Ron van Bekkum, Suze van Beusekom, and Robert Meinders for their help in recruiting participants;

Tessa, Iris, and Kirsten for transcribing the interviews.

## Author Contributions

**Conceptualization:** Edith Hagedoren—Meuwissen, Uta Roentgen, Sandra Zwakhalen, Loek van der Heide, Ramon Daniëls.

**Formal analysis:** Edith Hagedoren—Meuwissen, Uta Roentgen.

**Funding acquisition:** Edith Hagedoren—Meuwissen, Uta Roentgen, Loek van der Heide, Ramon Daniëls.

**Investigation:** Edith Hagedoren—Meuwissen.

**Methodology:** Edith Hagedoren—Meuwissen, Uta Roentgen, Sandra Zwakhalen, Ramon Daniëls.

**Supervision:** Uta Roentgen, Sandra Zwakhalen, Ramon Daniëls.

**Writing – original draft:** Edith Hagedoren—Meuwissen.

**Writing – review & editing:** Uta Roentgen, Sandra Zwakhalen, Loek van der Heide, Marie Josee van Rijn, Ramon Daniëls.

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
