## [Decision Letter · Decision Letter 0]

19 Sep 2024

PONE-D-24-27618The impact of wearing compression hosiery and the use of assistive products for donning and doffing: A descriptive qualitative study into user experiencesPLOS ONE

Dear Dr. Hagedoren - Meuwissen,

Thank you for submitting your manuscript to PLOS ONE. After careful consideration, we feel that it has merit but does not fully meet PLOS ONE’s publication criteria as it currently stands. Therefore, we invite you to submit a revised version of the manuscript that addresses the points raised during the review process.

Please consider the points raised by the reviewers, in particular to specify the measures (mmHg) of compression rather than 'levels', which vary between countries.  In addition, some further details on the conceptual models upon which your interview guide was based would be helpful for readers.

We look forward to receiving your revised manuscript.

Kind regards,

Kathleen Finlayson

Academic Editor

PLOS ONE

Journal Requirements:

1. When submitting your revision, we need you to address these additional requirements. Please ensure that your manuscript meets PLOS ONE's style requirements, including those for file naming. The PLOS ONE style templates can be found at https://journals.plos.org/plosone/s/file?id=wjVg/PLOSOne_formatting_sample_main_body.pdf and https://journals.plos.org/plosone/s/file?id=ba62/PLOSOne_formatting_sample_title_authors_affiliations.pdf 2. In the online submission form, you indicated that "The consent form which is based on the template of the Dutch CCMO did not request permission to share anonymized data publicly. The data of participant who gave permission on the consent form to use their data for further research can be obtained form the first author upon request. It is also worth mentioning that the data are only available in Dutch." All PLOS journals now require all data underlying the findings described in their manuscript to be freely available to other researchers, either 1. In a public repository, 2. Within the manuscript itself, or 3. Uploaded as supplementary information.This policy applies to all data except where public deposition would breach compliance with the protocol approved by your research ethics board. If your data cannot be made publicly available for ethical or legal reasons (e.g., public availability would compromise patient privacy), please explain your reasons on resubmission and your exemption request will be escalated for approval.

Reviewers' comments:

Reviewer's Responses to Questions

**Comments to the Author**

1. Is the manuscript technically sound, and do the data support the conclusions?

Reviewer #1: Yes

Reviewer #2: Yes

2. Has the statistical analysis been performed appropriately and rigorously? 

Reviewer #1: N/A

Reviewer #2: Yes

3. Have the authors made all data underlying the findings in their manuscript fully available?

Reviewer #1: Yes

Reviewer #2: Yes

4. Is the manuscript presented in an intelligible fashion and written in standard English?

Reviewer #1: Yes

Reviewer #2: Yes

5. Review Comments to the Author

Reviewer #1: Please see some corrections in the text and the sheet with recommendations.

Recommendations for manufacturers and public and private insurers are lacking at the end of the article. This complete survey need directives for paying organisms

Reviewer #2: This is a very good and useful work that addresses the issue of increasing the acceptability of compression hosiery by improving the means of putting it on. This work is relevant for manufacturers of compression hosiery, who must also take care of the means of putting it on, accessible to all categories of patients.

6. PLOS authors have the option to publish the peer review history of their article (what does this mean?). If published, this will include your full peer review and any attached files.

Reviewer #1: No

Reviewer #2: **Yes: **Vadim Bogachev

---

## [Author Response · Author response to Decision Letter 0]

22 Nov 2024

The following section will address the manner in which the feedback provided by the academic editor and reviewers has been incorporated into the text.

Response to the academic editor: 

The style requirements were subjected to a process of verification and subsequent modification where necessary. 

The reference list was subjected to a review, during which any necessary modifications were implemented. Our search did not identify any papers that had been retracted. 

Availability of the data: We have sought the approval of the Medical Ethical Review Committee (METC Z) and the Data Protection Officer of Zuyd University of Applied Science for the sharing of anonymized data in a public forum. The aforementioned bodies have granted permission for this data to be made publicly available, on the condition that it cannot be traced back to individuals. To ensure this, the participants' professions have been removed from the transcripts. Zuyd University will shortly be utilizing DataverseNL for this purpose. The DOI link will be provided as soon as DataverseNL is operational.

The interview guide was made available in English as supporting information.

Response to reviewer 1

The corrections proposed have been incorporated into the manuscript and are visible in the track changes version.

In response to your request, an image illustrating the various types of APD was incorporated into the text. 

The frameworks used to identify the topics in the interview guide were explained in text and illustrated with figures. 

The compression classes were specified with measure of compression in mmHg. 

In accordance with the advice recommendations for manufacturers and health care insurers have been added to the conclusion section of the manuscript. 

Response to reviewer 2

It is encouraging to hear that it has been rated as good and useful. Thank you.

---

## [Editor Report · Decision Letter 1]

4 Dec 2024

The impact of wearing compression hosiery and the use of assistive products for donning and doffing: A descriptive qualitative study into user experiences

PONE-D-24-27618R1

Dear Dr. Hagedoren - Meuwissen,

We’re pleased to inform you that your manuscript has been judged scientifically suitable for publication and will be formally accepted for publication once it meets all outstanding technical requirements.

Kind regards,

Kathleen Finlayson

Academic Editor

PLOS ONE

Additional Editor Comments (optional):

Thank you for addressing the reviewers' feedback, which enhances the clarity of your methods and sample characteristics.

Some final amendments would be useful, i.e.:

-Please consider a wording change for the phrase 'adults who HAVE to wear compression' - I suspect that actually they have been advised to wear compression, however they do not 'have to' wear compression, which is their choice.

- A suggestion, consider the wording of the name of your second theme 'wearing compression hosiery' - which does not give an indication to the reader of the meaning of what wearing compression hosiery was to your participants - a theme should indicate the meaning of your results e.g. looking at your results, the meaning of wearing compression hosiery to your participants seems to be both - health benefits yet discomfort from wearing compression ..
---

## [Editor Report · Acceptance letter]

10 Dec 2024

PONE-D-24-27618R1 

PLOS ONE

Dear Dr. Hagedoren - Meuwissen, 

I'm pleased to inform you that your manuscript has been deemed suitable for publication in PLOS ONE. Congratulations! Your manuscript is now being handed over to our production team.

Kind regards, 

on behalf of

Dr. Kathleen Finlayson 

Academic Editor

PLOS ONE